# Aging and Light Stress Result in Overlapping and Unique Gene Expression Changes in Photoreceptors

**DOI:** 10.3390/genes13020264

**Published:** 2022-01-29

**Authors:** Spencer E. Escobedo, Sarah C. Stanhope, Ziyu Dong, Vikki M. Weake

**Affiliations:** 1Department of Biochemistry, Purdue University, West Lafayette, IN 47907, USA; sescobe@purdue.edu (S.E.E.); sstanhop@purdue.edu (S.C.S.); dong314@purdue.edu (Z.D.); 2Purdue University Center for Cancer Research, Purdue University, West Lafayette, IN 47907, USA

**Keywords:** *Drosophila*, photoreceptor, aging, blue light, neuronal gene expression, retinal degeneration, light-induced stress, oxidative stress

## Abstract

Advanced age is one of the leading risk factors for vision loss and eye disease. Photoreceptors are the primary sensory neurons of the eye. The extended photoreceptor cell lifespan, in addition to its high metabolic needs due to phototransduction, makes it critical for these neurons to continually respond to the stresses associated with aging by mounting an appropriate gene expression response. Here, we sought to untangle the more general neuronal age-dependent transcriptional signature of photoreceptors with that induced by light stress. To do this, we aged flies or exposed them to various durations of blue light, followed by photoreceptor nuclei-specific transcriptome profiling. Using this approach, we identified genes that are both common and uniquely regulated by aging and light induced stress. Whereas both age and blue light induce expression of DNA repair genes and a neuronal-specific signature of death, both conditions result in downregulation of phototransduction. Interestingly, blue light uniquely induced genes that directly counteract the overactivation of the phototransduction signaling cascade. Lastly, unique gene expression changes in aging photoreceptors included the downregulation of genes involved in membrane potential homeostasis and mitochondrial function, as well as the upregulation of immune response genes. We propose that light stress contributes to the aging transcriptome of photoreceptors, but that there are also other environmental or intrinsic factors involved in age-associated photoreceptor gene expression signatures.

## 1. Introduction

Aging is characterized by a decline in organismal function. The free radical theory of aging [1], and more broadly characterized damage-based theories [2], suggest that the accumulation of oxidative damage, in combination with other sources, culminate in the decline of function and ultimate demise of an organism with advanced age. In the eye, visual function decreases with age in both humans and model organisms such as *Drosophila* melanogaster [3,4,5]. This decrease in function is associated with physiological changes in multiple different cell types and structures in the eye, including the cornea [6], vitreous [7] and retina [8,9,10]. In addition, age is a major risk-factor for many eye-associated diseases, such as cataract, age-related macular degeneration, glaucoma, and retinopathy [11]. There is also a strong environmental component to age-associated eye disease with factors such as diet, smoking, and sunlight exposure contributing to a higher risk in developing disease [12,13,14]. Understanding how aging and environmental stress impact specific cell types in the eye will help provide insight into how these factors interact and lead to the onset of age-associated ocular disease.

Although light is essential for vision, it also represents a considerable stress to cells in the eye. Photoreceptors, the primary light-sensing neurons in the eye, respond to light through the absorption of a photon by the molecule retinal contained within the G-coupled protein receptor Rhodopsin [15,16]. This process is highly conserved between *Drosophila* and humans, with rhodopsin activating the G-protein transducin, which leads to the downstream phototransduction cascade [15]. In flies, but not in vertebrates, Rhodopsin can be directly converted back to its inactive state within a single photoreceptor cell [17,18]. In outer photoreceptors (R1–R6 cells), blue light activates Rhodopsin 1 (Rh1) forming metaRhodopsin 1 (mRh1), which can be converted back to Rh1 through the absorption of orange light present in the spectra of normal white light [18,19]. Thus, in flies, prolonged exposure to blue light in the absence of other wavelengths results in constitutive activation of the phototransduction pathway. This overactivation of the phototransduction cascade results in excessive endocytosis of mRh1 and prolonged calcium influx into the photoreceptor that eventually results in its death [20,21,22,23,24]. We previously showed that the retinal degeneration caused by blue light exposure in *Drosophila* is caused by an increase in oxidative stress, particularly lipid peroxidation, that results from calcium excitotoxicity [20]. In aging flies, blue light exposure also increases markers of oxidative stress and decreases lifespan [25]. Thus, in *Drosophila*, blue light exposure provides a unique model in which to assess the impact of oxidative stress on the eye, with potential implications for aging within the organism.

During aging, there are changes in gene expression that lead to defined transcriptomic signatures that include aspects unique to particular tissues [26]. For example, distinct age-associated transcriptional signatures are present in specific cell types in the eye, including the human retina [27,28] and *Drosophila* photoreceptors [4]. We previously profiled the photoreceptor transcriptome of young flies exposed to blue light [29], identifying some similarities in the functional categories of genes that were differentially regulated during aging or under blue light stress. However, the different ages, genotypes, and eye color of the flies used confounded direct comparison of these datasets. Here, we determine the photoreceptor transcriptome in a single white-eyed fly strain during aging or after blue light exposure, allowing us to directly compare the gene expression changes upon either condition. We identify common gene expression signatures between light stress and aging suggesting that light stress is a major contributor to the aging photoreceptor transcriptome. However, we also observe unique gene expression signatures in aging photoreceptors, indicating that other factors also contribute to the transcriptional changes that occur in these aging cells independent of light. Unexpectedly, we also identify overexpression of an enzyme that counteracts oxidative stress in the white-eyed flies used in this study. We find that this genetic background is protected against retinal degeneration induced by blue light or aging, suggesting that increasing oxidative stress is the primary cause of photoreceptor degeneration under both conditions.

## 2. Materials and Methods

### 2.1. Fly Stocks, Husbandry, and Blue Light Stress

All experiments were conducted with male *Rh1-GFP^KASH^* flies in a *cn bw* background to deplete eye pigment [30]; these flies have white eyes. Two independent lines were used: *w*^1118^*; cn bw; P{w^+mC^ = Rh1-GFP-Msp300KASH}3-1* or *w*^1118^*; cn bw; P{w^+mC^ = Rh1-GFP-Msp300KASH}3–2*. To generate these fly stocks, the GFP-Msp300KASH cassette was excised from the pUASTattB-GFP-Msp300KASH vector [31] (Addgene #170806) using EcoRI and XbaI, and directionally cloned into pCaSpeR-ninaEp-GeneswitchGal4 [32] digested with the same enzymes to remove the GeneswitchGal4 cassette. Transgenic flies carrying *Rh1-GFP^KASH^* were generated by *P*-element mediated transformation into the *w*^1118^ strain. The chromosomal insertion site for *Rh1-GFP^KASH^* flies on chromosome 3 (line 3–1) was mapped using inverse PCR to 3L: 18,822,623, in the 5′ untranslated region (UTR) of the gene *Catalase*. Flies were raised on cornmeal agar food (6.07 g agar type 2, 32 g sugar, 50 g yeast, 50 g cornmeal, 3.2 g methyl paraben for preservation in 1.3 L water) at 25 °C and 65–75% humidity under a 12:12 light:dark cycle. The wavelength spectrum in the incubator was measured using a BLACK-Comet UV-VIS Spectrometer (StellarNet model #BLK-C) (Appendix A). Age-matched flies were collected on the first day after eclosion (day one; D1), and aged as described, transferring to fresh food every 2–3 days. All blue light experiments were conducted during the light cycle using a custom designed light stimulator [20]. Vials containing up to 50 day five (D5) flies were exposed to blue light (λ  =  465 nm) at 8000 lux (2 mW/cm^2^) for either three, five, or eight hours (h). For RNA-seq, flies were immediately flash frozen in liquid nitrogen post exposure. For phalloidin staining, flies were moved to a dark box in a 25 °C incubator and aged for seven days prior to processing.

### 2.2. Retina Staining and Analysis

Fly eyes were cut using microdissection scissors in phosphate buffered saline (PBS) with added detergent and fixative (0.1% Triton-X 100, 4% formaldehyde), leaving the corneal lens attached. Dissected eyes were then trimmed to remove excess brain tissue and cuticle, and fixed for a total of 20 min. Eyes were rinsed in PBST (0.1% Triton X-100) briefly, then transferred to 200 µL of PBST (0.1% Triton X-100,) containing 1:50 Phalloidin 594 (ThermoFisher, Waltham, MA, USA, #A12381) in a 0.5 mL microcentrifuge tube and incubated with rocking overnight at 4 °C. Following this, eyes were washed three times for five min each in 200 µL of PBST (0.1% Triton X-100). Eyes were then equilibrated in VectaShield (Vector Laboratories, Burlingame, CA, USA, #H100010) for 15 min and mounted lens side up on bridged slides using two #0 coverslips as bridges with a #1.5 coverslip. Samples were stored in the dark at 4 °C for up to one week prior to imaging. Confocal fluorescence microscopy was conducted with a Zeiss LSM880 confocal microscope using a Zeiss Plan-Apochromat 20X/.8 objective. Five or more biological replicates were imaged per condition. To quantify rhabdomere degeneration, all in-focus ommatidia in a single focal plane were scored for the presence of all seven visible rhabdomeres. Scores are shown as a percentage of intact rhabdomeres.

### 2.3. Nuclei Immunoprecipitation and RNA-Seq

R1–R6 photoreceptor-specific, GFP-labeled nuclei immunoprecipitation was conducted as previously described [33]. Between 150 and 250 male flies were used per sample, and three biological replicates were conducted for all RNA-seq experiments. Following nuclei immunoprecipitation, bead-bound nuclei were lysed in 300 µL of TRIzol (Zymo, Irvine, CA, USA, Cat # R2050) and RNA was extracted using the Direct-zol RNA MicroPrep kit (Zymo, Irvine, CA, USA, Cat # R2050) with DNase I treatment. RNA-seq libraries were constructed using the Universal RNA-Seq Library Preparation Kit with *Drosophila* specific rRNA depletion (Tecan Genomics, Zurich, CH, Cat# 0520-A01) using 10 ng of RNA to generate each library. Libraries were sequenced (150 bp, PE) using the Illumina HiSeq 4000 platform (Novogene). Between 40–80 million reads were obtained per sample.

### 2.4. Bioinformatics

RNA-seq reads were trimmed using Trimmomatic [34] (v0.39) to remove low quality reads >36 bp. High quality paired and unpaired reads were then aligned to the *Drosophila* genome (*Drosophila_melanogaster.BDGP6.28*) using HISAT2 [35] (v2.1.0). BAM files were generated and sorted using Samtools [36] (v1.8), and count files obtained using HTseq [37] (v0.13.5). Counts were filtered to remove lowly expressed genes (counts per million (cpm) > 10 in all samples), and normalized using the RUVs approach (k = 2) from RUVSeq [38] (v1.24.0), to obtain normalized cpms for 7929 expressed genes. Differential gene expression was performed using edgeR [39] (v3.32.1). Heatmaps of normalized expression (cpm) were generated using pheatmap [40] (v1.0.12). Enriched GO terms were identified and compared between samples using ClusterProfiler [41,42] (v3.18.1). Hypergeometric testing (Fisher’s exact tests) of significant pairwise intersections was conducted using SuperExactTest [43] (v1.0.7). Cnetplots were generated using ClusterProfiler. All plots were generated with R (v4.0.5) using custom scripts.

### 2.5. RT-qPCR from Whole Heads

Whole heads were dissected from five adult male flies, and RNA was extracted using Trizol, followed by Zymoprep Direct-zol RNA MicroPrep kit (Zymo Research, Zurich, Switzerland, Cat # R2050) with DNAase I treatment. cDNA was generated from 100 ng of RNA using Episcript Reverse Transcriptase (Epicentre, Petaluma, CA, USA) and random hexamer primers. RT-qPCR was conducted as previously described [44]. *Catalase* gene expression was normalized to the geometric mean of two reference genes (*RpL32*, *eIF1a*). Statistical significance was determined using Student’s *t*-test. 

### 2.6. Glutathione Redox Ratios in Dissected Eyes

Oxidized (GSSG) and reduced (GSH) glutathione levels were quantified in 25 dissected eyes per sample as described previously [45].

## 3. Results

### 3.1. Age and Blue Light Exposure Induce Global Changes in the Photoreceptor Transcriptome

We previously showed that prolonged exposure to blue light (8 h) induces retinal degeneration in young (day six, D6) white-eyed flies [46]. Retinal degeneration in flies is first apparent through the loss of the photoreceptor rhabdomere, which in most cases is followed by the degeneration of the remainder of the neuron [47]. Because other studies have also observed premature retinal degeneration in white-eyed flies (*w*^1118^) that were not exposed to any light stress [48], we sought to compare gene expression changes in flies exposed to light stress with flies undergoing normal aging. To examine the photoreceptor transcriptome, we used flies that express GFP^KASH^, which localizes to the outer nuclear membrane enabling immunoprecipitation of tagged nuclei [31,33]. GFP^KASH^ was expressed directly under control of the *ninaE* (Rh1) regulatory elements, which target expression specifically in the adult outer photoreceptors R1–R6. To generate white-eyed flies, which are sensitized to blue light stress [49], we crossed *Rh1-GFP^KASH^* into a *cn bw* background, which depletes eye pigment [30]. 

Next, we examined the photoreceptor transcriptome of flies exposed to light stress and compared this with normal aging. To do this, we exposed young D5 flies to blue light of varying durations (3, 5, or 8 h), or aged flies under standard 12:12 light:dark conditions to D5, D15, or D30, and isolated photoreceptor nuclei for RNA-seq under each condition (Figure 1A). We note that nuclear RNA provides a snapshot of actively transcribed RNA, rather than the steady-state mRNA levels in the cell that are measured using other RNA-seq approaches. To examine the differences between each condition, we performed principal component analysis (PCA) on the normalized counts (Figure 1B). The three biological replicates grouped together for each condition by PCA, separating by age along PC1 (28.07% variation), and by blue light exposure along both the PC1 and PC2 axes. These data suggest that there are both overlapping and distinct changes in gene expression in response to blue light or during aging. We note that PC1 and PC2 only explain approximately 50% of the variation, and there is additional variation present particularly between the aging samples, which is more apparent when the relative gene expression is plotted for each sample (Figure 1C,D).

To identify the genes with differential expression during aging or blue light exposure, we compared all conditions to the D5 untreated samples. Using this approach, we identified 1365, 957, and 1509 significantly differentially expressed genes (FDR < 0.05) in flies exposed to 3, 5, or 8 h blue light, respectively (Appendix A). We identified fewer genes that were differentially expressed during aging at D15 (226 genes), but this number increased to 1069 genes by D30 (Appendix A). We then compared differentially expressed gene sets between all the conditions and found that there were substantial overlapping sets of genes between many of the pair-wise comparisons (Figure 1C). Additionally, we identified 663 genes that were differentially expressed in three or more conditions (Figure 1C), suggesting that light stress and aging have some overlapping effects on the photoreceptor transcriptome. To further compare the global differences in gene expression, we first compiled a list of all 3453 genes that were differentially expressed in at least one condition. We then generated a heatmap showing the relative expression of each gene across all samples, clustering both samples and genes, and normalizing expression by row (gene) to highlight differences between the conditions (Figure 1D). Samples clustered together using this approach, similar to the PCA, and higher order clustering revealed similarities between the aging samples (D15 and D30) and the longest blue light exposure (8 h) (Figure 1D). However, there remain substantial differences between blue light and aging even at the longest blue light exposures, suggesting that there are distinct gene expression changes under both conditions. We conclude that shorter durations of exposure to blue light induce gene expression changes that may be more specific to light stress, whereas longer blue light exposure mimics aging. 

### 3.2. Stress-Response Genes Are Upregulated upon Blue Light Exposure or Aging, While Behavioral and Neuronal-Specific Genes Are Downregulated

Next, we examined the functional classes of genes that were differentially expressed during blue light exposure or aging. To do this, we first separated differentially expressed genes into those that were upregulated (increased expression upon blue light exposure or older age) or downregulated (decreased expression in either condition). We then performed Gene Ontology (GO) term analysis and compared the GO terms identified for each group to identify overlapping or similar functional sets of genes. In Figure 2, we show all significantly enriched GO terms (FDR < 0.05) for either upregulated (Figure 2A) or downregulated (Figure 2B) gene sets for each condition using dot plots, highlighting both shared and unique GO terms between blue light and aging conditions. Full annotated lists of all GO terms are provided in Appendix A. We did not identify any significantly enriched GO terms in the upregulated genes at 3 h blue light. However, there were substantial overlaps between blue light and aging GO terms in the upregulated genes, particularly upon 8 h blue light exposure for functions such as DNA repair, DNA metabolic process, and cell cycle (Figure 2A).

In contrast, aging samples were uniquely enriched for GO terms such as protein refolding and response to heat, whereas blue light uniquely enriched for GO terms associated with cell organization and development. When we examined the downregulated genes, we observed a larger overlap between blue light and age enriched GO terms compared with the upregulated genes (Figure 2B). GO terms such as axon guidance, trans-synaptic signaling, ion transport, and rhythmic behavior were common between blue light and aging in the downregulated genes. In contrast, blue light samples were uniquely enriched for GO terms such as eye morphogenesis, neuron differentiation and cell differentiation, while aging samples enriched for terms such as visual behavior and adrenergic receptor signaling. These data suggest that during aging and blue light treatment, photoreceptors induce expression of stress response pathways and metabolic genes, while genes related to neuronal function are downregulated. However, we also observed many unique GO terms in blue light versus aging, again suggesting that there are distinctive gene regulatory pathways specific to each condition.

### 3.3. The Flies Used in This Study Are Potentially Resistant to Light Stress Due to Overexpression of Catalase in Photoreceptors

We were surprised that there were no significantly enriched GO terms in the genes that were upregulated upon 3 h of blue light exposure (Figure 2A) because our previous analysis of gene expression in photoreceptors from similar aged flies exposed to blue light identified substantial enrichment of GO terms associated with stress response [29]. The enrichment of stress-response associated GO terms in [29] was largely driven by the strong upregulation of *Heat shock* genes, so we directly compared the induction of these genes in both studies (Figure 3A). In this comparison, the log_2_ fold change for each gene is determined relative to the appropriate control in each study (3 h dark treatment for [29] versus 0 h exposure in the current study). Unexpectedly, most of the stress response genes were not significantly induced in our study even though these were significantly upregulated in the previous study, despite the similarity in blue light treatment, age, and eye color of the flies used (Figure 3A). Because the EGFP used in the GFP^KASH^ nuclear membrane tag absorbs blue light in the 440–500 nm range with a λ_max_ of 488 nm [50], we initially wondered if the high levels of GFP^KASH^ protein present in the photoreceptors might diminish the negative impact of blue light on photoreceptor survival in the aging white-eyed flies, as well as our blue light exposure, which has a λ_max_ of 465 nm [20]. We note that the white LEDs present in the incubator used to raise these flies have a strong peak in the blue wavelengths (Appendix A), which is a common feature of commercially available white LEDs [51,52,53]. GFP-dependent photoprotection has been observed in reef coral experiencing high light stress [54,55,56]. However, since both white-eyed fly genotypes express photoreceptor-specific GFP, this is unlikely to underly the difference in sensitivity to blue light induced gene expression changes observed.

We next examined the genotype of the flies used in each study. In the previous study we used *cn bw; Rh1-Gal4* > *UAS-GFP^KASH^* flies [29], but for this study we developed new more efficiently expressed GFP^KASH^ transgenes that were directly under control of Rh1 genomic regulatory elements. These lines were used because they had higher nuclear RNA yields using the nuclei immuno-enrichment approach, and were created by *P*-element mediated transformation, generating insertions on multiple chromosomes to facilitate various genetic experiments. We carefully tested a subset of these lines to ensure that there was no premature age-dependent retinal degeneration in pigmented (red eye) backgrounds because insertion of the transgene could have resulted in mutation of a gene that was necessary for photoreceptor survival. When we examined the insertion site of the *Rh1-GFP^KASH^* transgene used in the current study (referred to hereafter as line 3–1), we found that it was present near the 5′ UTR of the *Catalase* gene. *Catalase* encodes an enzyme that reduces hydrogen peroxide (H_2_O_2_), counteracting oxidative stress, and is protective in aging models [57,58,59]. Since the *Rh1-GFP^KASH^ 3–1* flies are viable, and homozygous lines do not exhibit premature retinal degeneration, *Catalase* expression is unlikely to be disrupted in these flies. Instead, to our surprise, when we examined the expression level of *Catalase* in the RNA-seq analysis from the current study and compared this with previously published RNA-seq data using *Rh1-Gal4* > *UAS-GFP^KASH^* flies, we observed substantially higher levels of *Catalase* expression in all the *Rh1-GFP^KASH^ 3–1* samples relative to *Rh1-Gal4* > *UAS-GFP^KASH^* flies [4,29] (Figure 3B). We note that all RNA-seq samples compared in Figure 3B consist of total ribo-depleted nuclear RNA. To examine the steady-state mRNA levels of *Catalase* expression in the *Rh1-GFP^KASH^ 3–1* flies, we performed qRT-PCR analysis in two independent third chromosome insertions of *Rh1-GFP^KASH^*: lines 3–1 and 3–2, both in a *cn bw* background. We also observed significantly higher *Catalase* expression in line 3–1, which has the insertion in the 5′ region of *Catalase*, relative to line 3–2, but not to that same extent as observed in the nuclear RNA measurements. We conclude that insertion of the very strong Rh1 genomic regulatory elements near the 5′ UTR of *Catalase* results in potent induction of *Catalase* transcription in photoreceptor nuclei, which contributes to higher steady-state expression of *Catalase* mRNA. Thus, the white-eyed flies used in this study are likely protected against some of the oxidative stress associated with blue light or aging and should be considered an ameliorated model with regards to reactive oxygen species (ROS) levels. However, stress response genes are induced in the Rh1-GFP^KASH^ flies at older time points (D15, D30) and upon longer blue light exposures, suggesting that although these flies may be resistant at lower blue light exposures or earlier stages of aging, they do eventually experience the same types of gene expression changes observed in more sensitive models. We note that in some respects, these resistant flies provide advantages for studying gene expression changes associated with light stress versus aging because substantial retinal degeneration could interfere with our ability to obtain sufficient nuclear RNA and compare light stress and aging in the same flies.

### 3.4. Aging and Light Stress Have Unique and Overlapping Effects on Gene Expression in Photoreceptors

To examine the common and distinct pathways regulated by blue light and aging in more detail, we next directly compared the differentially expressed gene sets between each condition for either the upregulated or downregulated genes. We compared these groups separately because we reasoned that the direction of change in expression should be the same under blue light exposure or aging if these genes have a shared biological function in either process. As suggested by the GO term analysis in Figure 2, we observed a high overlap between the upregulated genes in both age and blue light (Figure 4A) that was significant for all pairwise comparisons (Figure 4B).

Although the pairwise comparisons between flies exposed to either blue light or during aging showed the most significant overlap (compare 3 h vs 5 h blue, or D15 vs D30), we also observed a highly significant overlap between 8 h blue light and D30. To examine these overlaps in more detail, we generated cnetplots that illustrate the common differentially expressed genes that contribute to each of the enriched GO terms. The cnetplot generated from the commonly upregulated genes identified under 8 h blue light and late aging (D30) revealed multiple GO terms related to DNA damage response and repair (Figure 4C). These GO terms were driven by upregulation of genes such as *Xpac* and *Rrp1*, which encode proteins that recognize and catalyze steps in base excision repair. Similarly, *Rad50* and *mre11*, two other genes commonly upregulated in 8 h of blue light and D30, encode the components of the DNA damage repair (DDR) complex involved in recognition and repair of double stranded breaks. We also observed GO terms not directly associated with the DNA damage response that were related to the cell cycle. Previous studies have identified the upregulation of cell cycle genes in post-mitotic neurons undergoing programmed cell death in both age-related disease [60] or DNA damage models [61,62,63], suggesting the similar upregulation we observe in aged photoreceptors may also represent a neuronal cell death transcriptional signature. While blue light and aging share common upregulated genes, aging flies at D15 and D30 also exhibited shared upregulation of genes associated with DNA damage and stress response (Figure 4D). Interestingly, many of the same genes commonly upregulated with age (Figure 4D) were also identified in the overlap between 8 h and D30 (Figure 4C), suggesting that both blue light and aging induce similar upregulation of genes involved in the DNA damage response. Although a significant number of genes were differentially expressed in both 3 h and D30 (85 genes), as well as 5 h and D30 (54 genes), no GO terms were significantly enriched in these shared gene sets.

While significant overlaps exist between upregulated genes in aging and blue light exposed flies, many unique gene expression changes also occur under each condition. To explore the differences between age and blue light, we focused on the uniquely upregulated genes in either D30 (368 genes) or 8 h blue light (568 genes) samples. GO term analysis of genes that were uniquely upregulated at D30 (age) enriched for processes involved in the defense and immune response, rRNA processing, ribosome biogenesis, and nucleoside metabolism (Appendix A). The upregulation of immune and defense response pathways may signal that aging flies have an accumulation of pathogens [64,65], while the increase in ribosome biogenesis indicates a regulatory feedback loop from the disruption or dysregulation in protein synthesis [66]. In contrast, 8 h blue light resulted in the unique upregulation of a large number of genes involved in cellular organization, growth, neuronal projections, nucleotide excision repair (NER), and the negative regulation of response to stimulus (Appendix A). For example, *PKa-R1* and *Vps33B* are involved in suppressing the light response [67,68], suggest that photoreceptors induce expression of genes to directly counteract the overactivation of the phototransduction signaling cascade caused by blue light exposure, and that this response is not induced during aging.

We also observed a high overlap of downregulated genes between aging and blue light (Figure 5A), which were significant for all pairwise comparisons (Figure 5B). While the most significant pairwise comparisons were between blue light and aging conditions, the most significant overlap between blue light and age was between 8 h and D30. The cnetplot of this overlapping gene set revealed genes associated with phototransduction such as *shakB* and *trpl*. (Figure 5C). When we evaluated the overlap between D15 and D30, we identified genes associated with neurotransmitter signaling and behavior (Figure 5D). For example, *Gabat* and *AANAT1* both encode enzymes that modify neurotransmitters involved in the regulation of sleep. Based on this, we conclude that extended blue light exposure (8 h) induces similar changes in phototransduction genes as those in old D30 photoreceptors.

Last, we asked what unique processes were downregulated with age or blue light. We identified 190 and 287 unique downregulated genes in D30 and 8 h blue light samples, respectively. Surprisingly, no GO terms were enriched in the uniquely downregulated genes upon 8 h of blue light. However, downregulated genes specific to D30 enriched for functions related to cation and transmembrane ion transport (Appendix A). Specifically, these genes encode multiple potassium channels, such as *KCNQ*, *kcc*, *CG3078*, and *Ork1*. Potassium transport is an important stabilizer of photoreceptor membrane potential [69,70], and the downregulation of these genes suggests that this process may become dysregulated in aging photoreceptors. Additionally, we observed age-specific downregulation of genes involved in ATP synthesis, such as *ATPsynb*, *blw*, and *VhaM9.7-a*. This suggests that that there might be a decrease in energy production in aging photoreceptors that is independent of light stress.

### 3.5. Catalase Overexpression Correlates with Decreased Retinal Degeneration and Oxidative Stress Levels

Because the white-eyed *Rh1-GFP^KASH^ 3–1* flies used in the current study exhibited high levels of *Catalase* expression in photoreceptors, and delayed gene expression responses relative to previous studies [20,45], we wondered if these flies would also be protected against the retinal degeneration induced by blue light exposure in white-eyed flies. Prolonged exposure to blue light (8 h) induces retinal degeneration in young (day six, D6) white-eyed flies, although there are differences in severity between *w*^1118^ and *cn bw* genotypes [45]. We assessed retinal degeneration in our white-eyed *Rh1-GFP^KASH^ 3–1* flies during aging or light stress (see methods). To do this, we exposed young D5 flies to blue light of varying durations (3, 5, or 8 h), or aged flies under standard 12:12 light:dark conditions to time points between D5 and D50 (D5, D15, D30, D50), and assessed retinal degeneration (rhabdomere loss) by staining retinas with phalloidin, which marks the actin-rich rhabdomeres in the photoreceptors (Figure 6A). Similar to our previous observations for white-eyed flies expressing GFP^KASH^ [29], shorter durations of blue light exposure (3–5 h) did not induce substantial retinal degeneration (Figure 6B). Interestingly, prolonged blue light exposure (8 h) also did not result in significant rhabdomere loss; however, multiple replicates showed a degeneration phenotype, albeit to a lesser extent than that observed previously in *w*^1118^ or *cn bw* flies that do not express GFP [45]. Moreover, in contrast to observations for *w*^1118^ flies, which show initial signs of retinal degeneration by D5 that progress to substantial rhabdomere loss at D15 and D30 [47], we did not observe significant retinal degeneration with age in the genotype used for this study (Figure 6C). However, similar to 8 h blue light exposure, multiple D50 replicates of line 3-1 showed a retinal degeneration phenotype.

To test if the overexpression of *Catalase* in the *Rh1-GFP^KASH^ 3–1* flies correlated with protection from blue light-induced retinal degeneration, we assessed blue light-induced retinal degeneration in the *Rh1-GFP^KASH^ 3–2* flies that did not overexpress *Catalase* and in *w*^1118^ flies using an identical approach to that used for the 3–1 line. In contrast to line 3–1, we did observe significant rhabdomere loss in the *Rh1-GFP^KASH^ 3*–*2* line and in *w*^1118^ flies of the same age exposed to 8 h blue light exposure, consistent with previous studies [20,45] (Figure 6B). These data indicate that photoreceptor overexpression of *Catalase* in the white-eyed flies used in the current study (line 3–1) correlates with decreased retinal degeneration resulting from prolonged (8 h) blue light treatment. Since we did observe early sporadic signs of rhabdomere loss in the flies used for this study both at 8 h and at older time points (D50), we predict that retinal degeneration might only be delayed in this genetic background relative to other white-eyed flies.

Oxidative stress increases in many tissues during aging [71,72], and we previously showed that blue light increases levels of H_2_O_2_ and a marker of lipid peroxidation, malondialdehyde, in the eye [46]. To compare oxidative stress induced by aging or blue light exposure in the flies used for this study relative to *w*^1118^, we examined ratios of reduced and oxidized glutathione (GSH:GSSG) in dissected eyes from flies treated as outlined in Figure 6A. Glutathione is an important thiol antioxidant that prevents the accumulation of toxic lipid peroxides [73,74]. The ratio of GSH:GSSG correlates with, and provides an indication of, the relative level of cellular oxidative stress [75,76]. In this assay, a decreased ratio of GSH:GSSG indicates higher levels of the oxidized glutathione pool, correlating with increased oxidative stress. Using this approach, we found a significant increase in oxidative stress in *w*^1118^ flies exposed to 8 h blue light but not in *cn bw; Rh1-GFP^KASH^ 3–1* (Figure 7A). However, we did observe a significant increase in oxidative stress in the latter genotype in D30 relative to D5 (Figure 7B). Surprisingly, there was a significant increase in the GSH:GSSG ratio in D15 flies relative to untreated D5, indicating that these flies have a more reduced glutathione pool which may infer a higher capacity to respond to oxidative damage than that of D5 flies, potentially correlating with *Catalase* expression, although this has not been tested in the current study. These data suggest that the early changes in aging white-eyed flies by D15 might include an increased antioxidant response, but that this is insufficient to counteract the high levels of oxidative stress present at older ages such as D30. These glutathione pool measurements, together with the retinal degeneration assessment in the fly line used in the current study, support a neuroprotective role for *Catalase* overexpression in photoreceptors in both light stress and aging models.

## 4. Discussion

Here, we directly compare gene expression changes that result from light stress or aging in photoreceptors, allowing us to separate the age-associated changes in gene expression caused by environmental light stress from those caused by other factors. Similar to our previous study in young (D6) white-eyed flies exposed to 3 h blue light [29], we also identified similar functional categories of genes that were differentially expressed upon different blue light exposure durations in these flies (D5). In particular, blue light upregulates expression of several stress response genes, indicating that this light exposure causes a substantial stress response in the fly photoreceptors. Other studies have shown that flies exposed to lower intensities of blue light throughout their lifespan exhibit increased expression of heat shock and oxidative stress genes [25]. When we compared blue light and aging, we identified substantial overlaps in gene expression between the longest blue light exposure condition and D30. However, oxidative stress levels as measured by oxidized:reduced glutathione levels were only significantly higher at D30, and not under 8 h blue light exposure. We previously observed increased H_2_O_2_ and malondialdehyde (lipid peroxidation) levels in eyes from *w*^1118^ flies exposed to 8 h blue light [46], indicating that blue light increases oxidative stress levels in the eye. Moreover, both these markers of oxidative stress and retinal degeneration were rescued by overexpression of *Cytochrome-b5*, which suppresses the lipid peroxidation resulting from blue light exposure [46]. We attribute the differences in severity/onset of the retinal degeneration in the flies used in the current study (*Rh1-GFP^KASH^ 3–1*) versus the previous lines used (*w*^1118^ or *cn bw*) to the high levels of *Catalase* expressed in photoreceptors resulting from the insertion position of the transgene. Recent studies have shown that *w*^1118^ (null mutation in the gene *white*) flies that lack eye pigment exhibit significant retinal degeneration in 50% of ommatidia by D5, and that retinal degeneration progresses further by D15 and D30 [48]. Despite this potential protective effect, the increase in stress response genes observed in our D5 flies exposed to 3, 5, or 8 h blue light suggests that they are experiencing oxidative stress, but that the overexpression of *Catalase*, which reduces H_2_O_2_ and decreases oxidative stress, might be sufficient to prevent global changes in glutathione ratios in the eye. This is particularly striking given that the overexpression of *Catalase* at steady-state level is relatively modest (~30% higher than control), even though *Catalase* nuclear transcript levels were much higher. Similarly, the higher reduced pool of glutathione observed in white-eyed D15 flies suggests that relatively young flies may be able to counteract the early effects of oxidative stress in part by increasing antioxidant capacity. It is interesting that *Catalase* expression correlates with protection against blue light stress because overexpression of *Superoxide dismutase 1* (*Sod1*) decreases H_2_O_2_ levels in the eye but does not protect against blue light-induced retinal degeneration [46]. Although the high level of *Catalase* expression in the *Rh1-GFP^KASH^ 3–1* flies generated in this study was inadvertent, these flies provide an intriguing model for further study to characterize the oxidative stress-dependent transcriptome of aging photoreceptors. The differences in level of *Catalase* expression observed between the photoreceptor nuclear RNA and whole head steady-state mRNA levels are also notable, suggesting that *Catalase* mRNA levels might be tightly regulated in these cells.

Overall, our data indicate that 15–25 percent of the gene expression changes observed in aging photoreceptors can likely be attributed to light stress. These light stress-dependent gene expression changes include the upregulation of stress response and DNA repair pathways, as well as a conserved neuronal cell death signature consisting of cell cycle genes. In addition, light stress appears to be responsible for the downregulation of phototransduction genes observed in aging photoreceptors. Although the current studies were performed in white-eyed flies, some similar GO terms were enriched when gene expression changes were analyzed during aging in flies with red eyes (*Rh1-Gal4* > *UAS-GFP^KASH^*), albeit at much later time points [4,33]. However, because of the differences in genetic background and nuclei immuno-enrichment technique used, it is not possible to directly compare the data from these studies. Thus, our data suggest that the eye ages more rapidly in white versus red-eyed flies, likely due to an increase in light stress dependent accumulation of damage that is ameliorated by the presence of neuroprotective pigments in red-eyed flies.

Although many pathways were commonly regulated by light stress and aging, we also identified gene expression signatures that were unique to each condition. During aging, genes involved in defense and immune response were upregulated while genes involved in ion transport and ATP synthesis were downregulated. These data indicate that aging photoreceptors might be exposed to other environmental stresses such as cellular damage or pathogens that induce immune response. Moreover, aspects of metabolism may become dysregulated in aging photoreceptors independent of light stress. Recent work by our lab has identified metabolic changes in the aging *Drosophila* eye that include changes to folate, S-adenosyl-methionine (SAM), and glutamate biosynthesis [77]. In contrast, blue light exposure upregulated genes involved in cellular organization and the negative regulation of stimulus, suggesting that blue light exposed photoreceptors induce gene expression pathways to quench the light response, likely as a neuroprotective mechanism. Since these pathways were not induced in aging photoreceptors, they likely represent a specialized response to light stress in the eye.

Together, our data support a model in which light stress and other factors contribute to the aging transcriptome of photoreceptors. Moreover, we demonstrate that white-eyed flies might provide a suitable model for premature aging in the *Drosophila* eye. Our study helps to define the complicated environmental and intrinsic signals that lead to changes in gene expression within a single cell type, photoreceptor neurons, in an aging organism.

## Figures and Tables

**Figure 1 genes-13-00264-f001:**
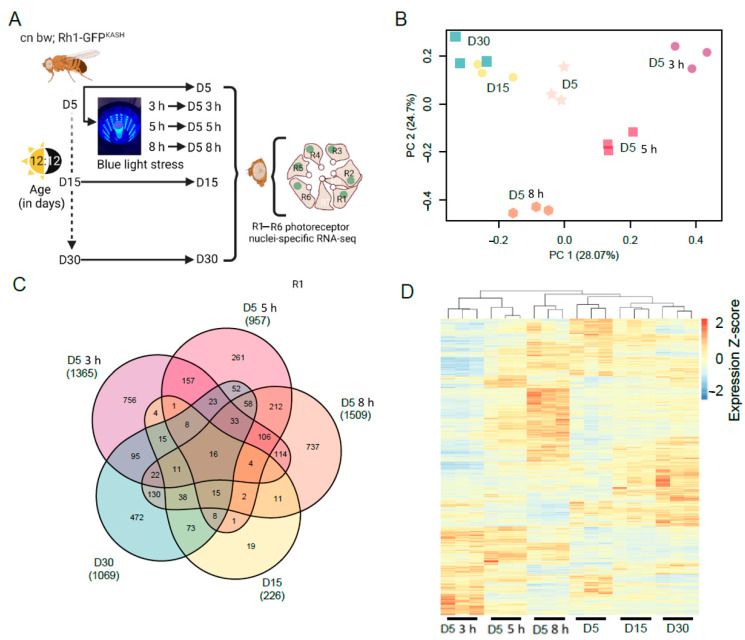
Global gene expression changes in white-eyed flies with age or blue light exposure. (**A**) Overview of the experimental design for photoreceptor nuclei-specific RNA-seq from blue light exposed or aged flies. Male white-eyed *Rh1-GFP^KASH^* flies at D5, D15 or D30, and D5 flies exposed to 3, 5, or 8 h of blue light were collected and photoreceptor nuclei-specific RNA-seq performed (*n* = 3). (**B**) PCA analysis of aged or blue light exposed flies. (**C**) Venn diagram of the differentially expressed genes in aged or blue light exposed flies identified relative to D5 untreated. The total number of significantly differentially expressed genes is shown under each sample name in brackets. (**D**) Hierarchical clustering of all differentially expressed genes in aged or blue light exposed flies. Normalized expression (Z-scores) were calculated for each gene (row).

**Figure 2 genes-13-00264-f002:**
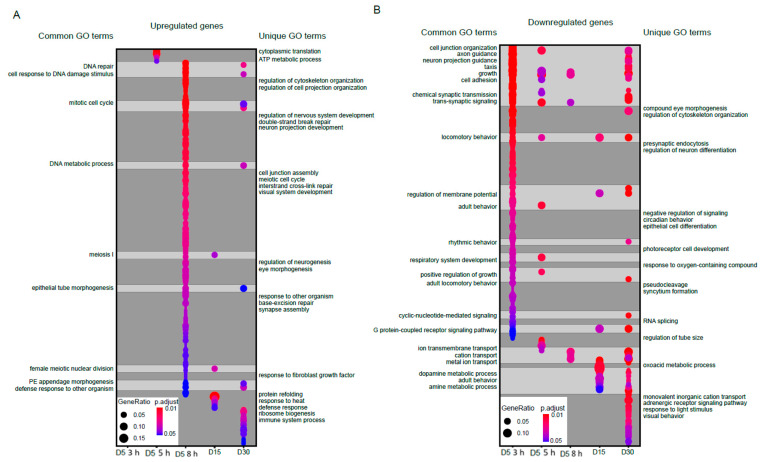
Significantly enriched GO terms for upregulated or downregulated genes. Dot plots of all enriched Gene Ontology (GO) terms for upregulated (**A**) or downregulated (**B**) genes in aged or blue light exposed flies relative to D5. Plots are shaded to highlight common (light grey) and unique (dark grey) GO terms between samples. Select GO terms of interest are annotated. A full list of GO terms is provided in Appendix A.

**Figure 3 genes-13-00264-f003:**
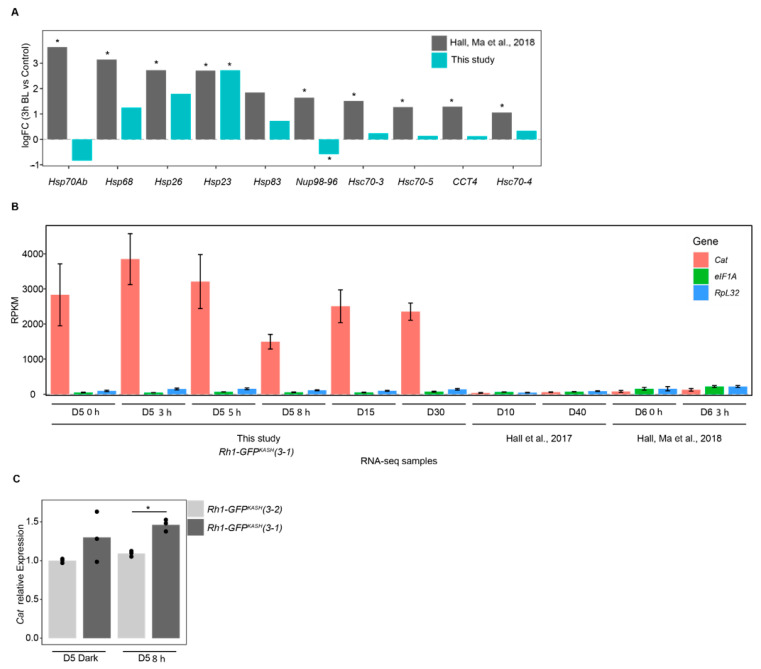
Overexpression of *Catalase* in *Rh1-GFP^KASH^* flies correlates with lower induction of stress-response genes following blue light exposure. (**A**) log_2_ fold-change values for 3 h blue light versus control for the indicated genes in Hall, Ma et al. 2018 versus this study. *, FDR < 0.05. (**B**) Relative expression (RPKM) of the indicated genes in samples from this study relative to Hall, Ma et al., 2018 and Hall et al., 2017. Panels A and B show nuclear transcript levels based on RNA-seq analysis. Age and/or blue light treatment indicated on *x* axis (D, days; h, blue light exposure time). Bars indicate mean ± SD. (**C**) Bar plots showing qPCR analysis of *Catalase* mRNA levels from male heads, representing steady-state transcript levels. Bars represent mean expression relative to reference genes with individual replicates overlaid as points. *, *p*-value < 0.005.

**Figure 4 genes-13-00264-f004:**
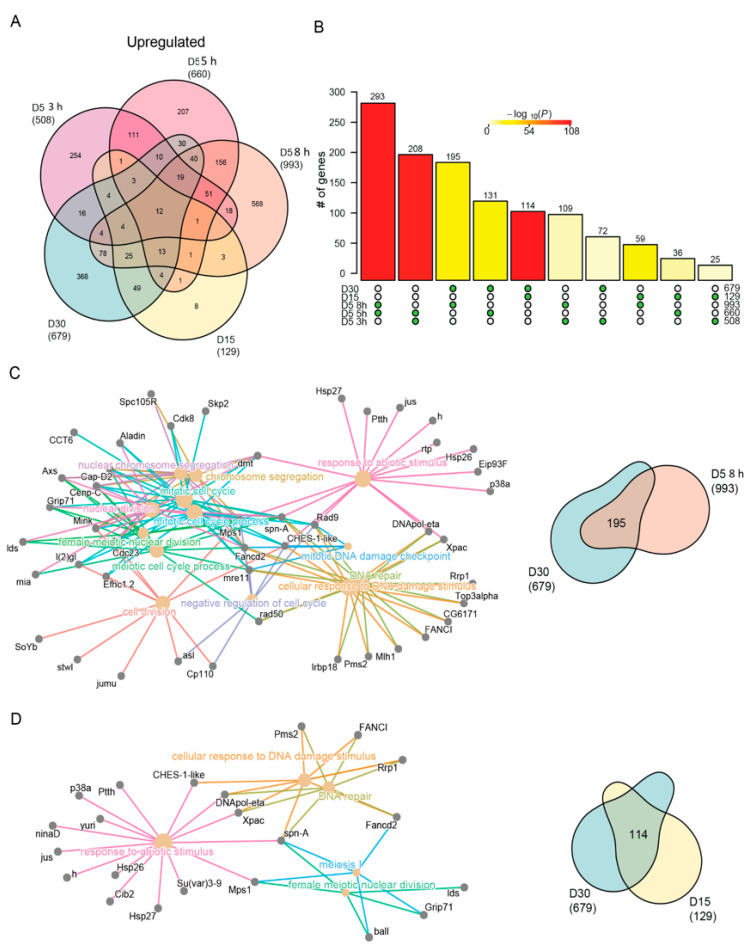
Similar stress response genes are induced with age and blue light stress. (**A**) Venn diagram of the differentially upregulated genes in aged or blue light exposed flies relative to D5. (**B**) Statistical analysis for all pairwise comparisons of significantly upregulated genes in aging or blue light exposed conditions. Bar height represents the number of genes in each pairwise overlap (green dots). Bars are colored by *p*-values from fisher’s exact test, with the null hypothesis being an overlap no greater than that expected by chance. Cnetplots representing the enriched GO terms and corresponding genes for the intersection between (**C**) 8 h and D30 or (**D**) D15 and D30 upregulated genes.

**Figure 5 genes-13-00264-f005:**
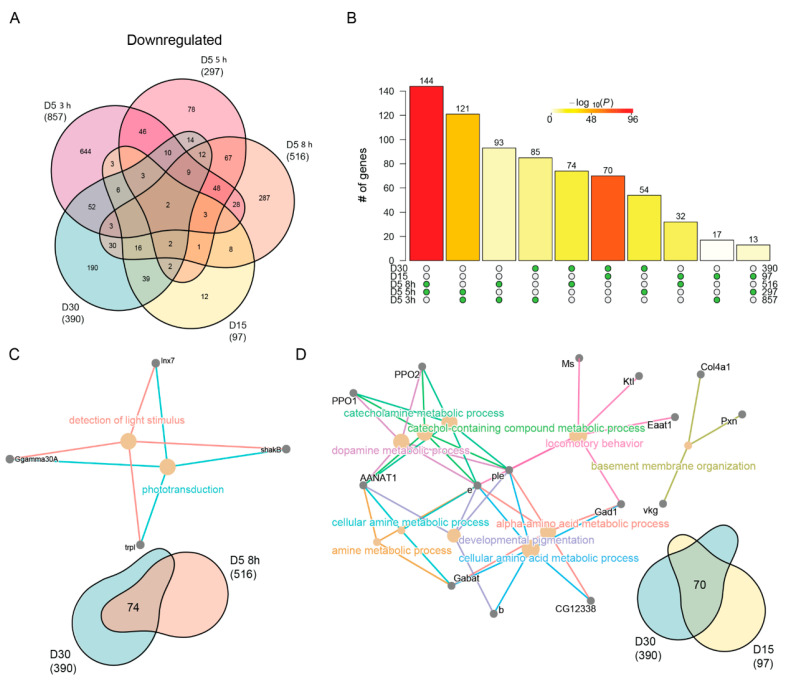
Neuronal specific genes are downregulated with age and blue light stress. (**A**) Venn diagram of the differentially downregulated genes in aged or blue light exposed flies relative to D5. (**B**) Statistical analysis for all pairwise comparisons of significantly downregulated genes in aging or blue light exposed conditions. Bar height represents the number of genes in each pairwise overlap (green dots). Bars are colored by *p*-values from fisher’s exact test, with the null hypothesis being an overlap no greater than that expected by chance. Cnetplots representing the enriched GO terms and corresponding genes for the intersection between (**C**) 8 h and D30 or (**D**) D15 and D30.

**Figure 6 genes-13-00264-f006:**
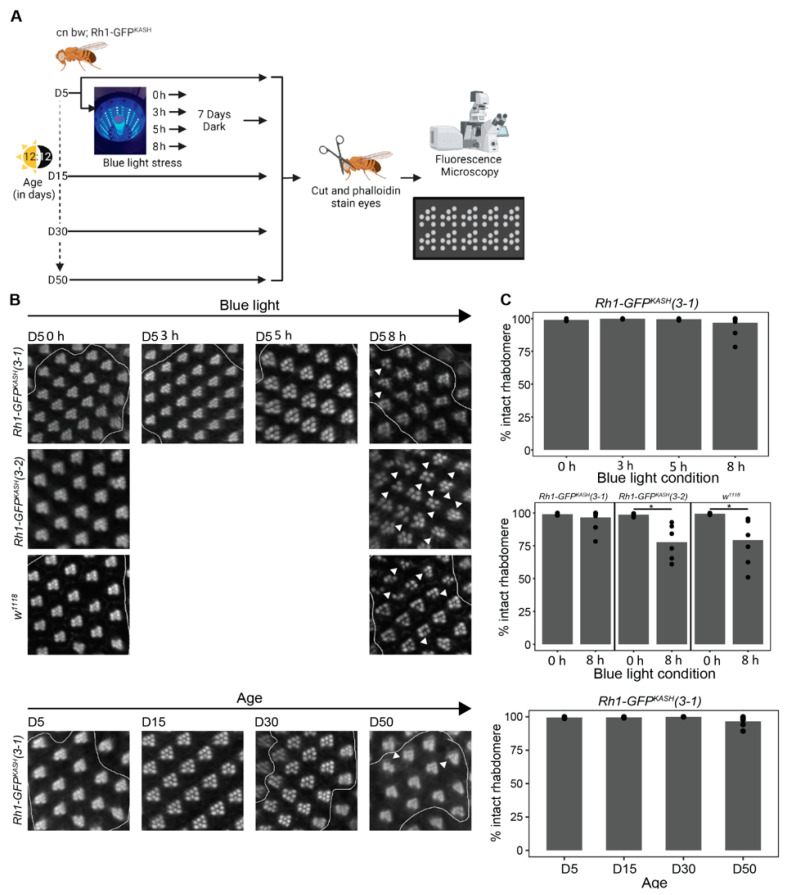
*Catalase* overexpression correlates with mild suppression of retinal degeneration in *Rh1-GFP^KASH^ (3–1)* flies. (**A**) Overview of the experimental design for assessing retinal degeneration in male flies of the indicated genotypes at D5, D15, D30 and D50, or D5 flies exposed to 0, 3, 5 or 8 h of blue light. (**B**) Representative images of dissected and phalloidin (F-actin) stained blue light exposed or aged eyes from the indicated genotypes (*cn bw; Rh1-GFP^KASH^ (3–1)*, *cn bw; Rh1-GFP^KASH^ (3–2)* or *w*^1118^). Arrow heads point to ommatidia or regions in the eye with degenerated rhabdomeres. (**C**) Quantification (displayed as percent of intact rhabdomere) of rhabdomere loss in aged or blue light exposed flies from the indicated genotypes (*n* ≥ 5). Black circles indicate individual biological replicates (flies), and mean shown by bar height. *p*-value (* <0.05), *t*-test.

**Figure 7 genes-13-00264-f007:**
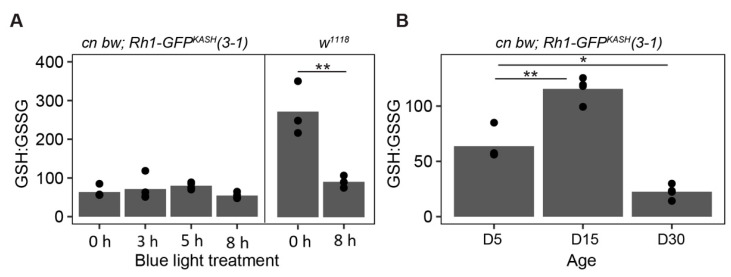
Lower levels of glutathione oxidation following blue light exposure in *Rh1-GFP^KASH^ 3–1* flies relative to *w*^1118^. (**A**) GSH:GSSG ratios in male white-eyed *cn bw Rh1-GFP^KASH^ 3–1* or *w*^1118^ flies eyes exposed to 3, 5 or 8 h of blue light at D5 versus untreated control (0 h). Black circles indicate individual biological replicates (*n* ≥ 3; 25 eyes per sample), and mean shown by bar height. *p*-value (** <0.005), *t*-test. (**B**) GSH:GSSG ratio in male white-eyed *cn bw Rh1-GFP^KASH^ 3–1* flies aged to D5, D15, or D30. *p*-value (* <0.05, ** <0.005), ANOVA with post-hoc.

## Data Availability

RNA-seq data are available from Gene Expression Omnibus (GEO) under accession GSE184389. Data were compared with previous RNA-seq data from our lab available under accessions GSE106820 and GSE83431.

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
