# Peer review of "Aging and Light Stress Result in Overlapping and Unique Gene Expression Changes in Photoreceptors"

_genes, 2022, doi:10.3390/genes13020264_

Round 1

Reviewer 1 Report

Retinal degenerative diseases are among the most common causes of vision loss, affecting
millions of people around the world. Numerous studies have shown that both age and light
exposure hasten retinal degeneration and vision loss. Thus, further understanding the molecular
pathways associated with these stressors are needed for identifying and developing effective
therapeutic targets. The submitted manuscript by Escobedo et al builds on previous studies by
the Weake group aimed at identifying such pathways, using the genetically-amenable model
system, Drosophila melanogaster. In particular, the current work establishes a
pigment-free/light-sensitive system to directly compare events associated with acute blue light
retinal damage and a more physiological, yet slower age-related retinal degeneration model,
using a photoreceptor-specific transcriptomic approach. The study is well-crafted and written,
and is carefully performed and analyzed, leading to the identification of common and unique
pathways affected by acute light damage vs age. Most notably, the authors find significant
overlap between gene expression changes that occur late in both models (8h blue light, and 30d
age), supporting the underlying hypothesis that the acute model serves as an effective proxy for
understanding slower, age-related degeneration events. The study design and experimental
outcomes will no doubt serve as an important framework for future mechanistic and comparative
studies.
Despite the valiant efforts by the authors to create a pigment-free model that should cause the
ell-animal to be more sensitive to light damage, they were faced with a perplexing result -
introduction of the transgene used for doing the photoreceptor-restricted RNA isolation led to
significantly less degeneration than previously reported by both this group as well as others.
Indeed, in both models, no significant loss in photoreceptor integrity was observed. In addition,
the expected change in oxidative stress previously observed in the blue light model was unable
to be reproduced in this new model. Thus, it is difficult to assess how well these studies can be
compared to others, and how this data should be interpreted with respect to retinal
degeneration. I have included a few ideas to consider to address this complication.

a. The authors have previously used phototaxis assays to quantify loss in visual function,
which they showed occurred even in the absence of morphologically-detectable
photoreceptor degeneration. Doing so in this study may also be an effective means to
compare and contrast the different models/time points in this study and provide a basis for
comparison to other work. Additionally, such experiments could be a nice test of the GO
term analysis, as the D30 predicts changes in visual behaviour while the blue light
treatment appears more related to morphogenesis.

b. The authors kindly provided the insertion site of the transgene. Surprisingly, it is located
directly with the 5’ end of the Catalase gene, a gene responsible for oxidative stress, a
pathway stressed by the authors in the introduction as an important contributor to the
aging process. One might expect this insertion to down-regulate Cat expression, thereby
making the photoreceptors more sensitive to damage, which is clearly not the case. An
alternative hypothesis could be that the Rh1 enhancer included in the construct (or the
insertion itself) may inappropriately increase Cat expression. Either way, it seems
appropriate to compare Cat gene expression in the flies used in this study vs that found in
flies used in previous work. Also, given the fact that the line used for the study does
interrupt an important gene in aging, were multiple insertion sites initially analyzed as candidate lines for the current studies, perhaps using the faster blue light model as a
system? If so, such information should be included in the Methods.

c. With respect to the limited degeneration observed in the blue light-treated animals, this
reviewer noted that degeneration was monitored after 12 days in the dark in earlier
studies, while the current study waited only 7 days - is it possible that this additional time
could account for the differences?

d. Finally, since diet and light levels influence retinal degeneration rates across the animal
kingdom, including Drosophila, more details of these factors should be included in the
Methods.

Additional editorial comments:
The last sentence of the abstract proposes a light-independent mechanism for photoreceptor
aging, but the last paragraph of the Discussion states that you data support a model in which
light stress does contribute to photoreceptor aging. Please clarify.
Fig 1B legend, p. 4, line 18: I do not see the arrow heads indicating degeneration
p. 8, line 259: states that the higher order clustering revealed similarities between the aging
samples and 8h blue light. While there are a few genes in the 8h sample that are in the same
vicinity as the D15/30 samples, it looks like most of the 8 hr genes cluster in a separate area. I
would try to re-word this more carefully.
p. 15, line 422: it reads as if you found upregulation of stress, DNA repair, AND cell death/cell
cycle genes in your previous work. However, in my re-reading of those papers, I wasn’t able to
find evidence for the cell cycle/cell death changes. Please clarify. Perhaps you could include a
supplemental file that directly compares the genes identified in the past and present work as
further support for the value of the current study moving forward.

Reviewer 2 Report

In the study titled with “Aging and light stress result in overlapping and unique gene expression changes in photoreceptors” by Spencer E Escobedoet al. reported the RNA-seq in photoreceptors of flies of different ages and under various length of blue light treatment. With bioinformatic analysis, they identified some commonly regulated genes and also several unique differentially expressed genes. While there are several weak points in the research and the manuscript should be further improved or modified. Please improve or modify all the weak points or the manuscript.

1)In figure 1 and section 3.1, the author found that there are some phenotypic differences between Rh1-GFPKASH and other white eyes flies they tested before, particularly they did not observe the significant retinal degeneration after 8 hrs of blue light treatment. The author proposed that there is GFP-dependent photoprotective mechanism involved. The author should provide some evidences to support this idea. This phenotype could be induced by the Rh1-GFPKASH insertion effect by P-element. The author should test this in another Rh1-GFPKASH flies which the transgene cassette was inserted at other genomic locus.

2)The data from figure 2 is very inconclusive. The GSH:GSSG ratios in blue light treated flies somehow matched the trend in figure 1C but still not match with previously treated other white eyes flies. At the same time, the data from different ages of flies is very hard to digest. First, the author should include the GSH:GSSG ratio of D50 flies. Second, the author’s explanations for the significant decrease of oxidative stress at D30 is not very persuasive. It is impossible to make a conclusion that oxidative stresses increase in the aging eye based on the current presented data. The author should provide more convincing data.

3) Based on the PCA analysis in figure 3B, there is no significant difference between libraries of D15 and D30 flies as they can’t be separated by either PC1 or PC2. This will make the following analysis of aging effects untrustable. The author should explain this in the context.

4)The author claims that “shorter durations of exposure to blue light induce gene expression changes that may be more specific to light stress, whereas longer blue light exposure mimics aging.” The author should provide some statistic data to support this conclusion. The author could compare the differentially expressed gene list in 3 or 5 hrs blue light treatment libraries with aging libraries and add some descriptions in section 3.3.

5) From figure 5-7, the author analyzed and presented either commonly up/downregulated or uniquely expressed genes from blue light treated and aging flies with Cnetplot. The author discussed some of the key pathways in the context, while there are too much gene names and GO terms in the figure which is very hard to capture important genes/pathways the author described. I’d like to recommend the author to highlight these key genes/pathways in the Cnetplot which will make the data more friendly to readers, especially in the figure 6.

6) One important supporting data missing in the manuscript is the validation or confirmation of RNA-seq with RT-PCR or other methods. The author should pick several interesting genes which described in the manuscript and verify their expression change with RT-PCR.

Minor Points:

1)Line 160-161, “Drosophila genotypes showed that while 3h of blue light did not result 160 in retinal degeneration29, 8h exposure caused significant degeneration45.” should be “Drosophila genotypes showed that 3h of blue light did not result 160 in retinal degeneration29, while 8h exposure caused significant degeneration45.”

Round 2

Reviewer 2 Report

The responses from the authors regarding my questions and concerns are reasonable. I understand the state from the authors. I still hope the author could perform the additional analysis and experiments I required.

Author Response

Please see as attachment.
